# SAGE: Sufficiency-Aware Implicit Graph Exploration for Long Context Reasoning

## Abstract

Large Language Models (LLMs) have achieved impressive progress in natural language processing, but their limited ability to retain long-term context constrains performance on document-level or multi-turn tasks. Retrieval-Augmented Generation (RAG) mitigates this by retrieving relevant information from an external corpus. Recently, graph-based RAG systems have shown promise for long-context reasoning. However, these methods face some challenges: high preprocessing computational costs, static graph architectures with fixed node and edge semantics, and complex parameter finetuning requirements that could limit their practical adoption. Recognizing that modern LLMs possess substantially improved reasoning capabilities, we propose SAGE, a dynamic implicit graph exploration framework that eliminates the need for explicit graph construction while preserving multi-hop reasoning benefits. Experiments are conducted on challenging long-context QA benchmarks, including NovelQA and Marathon. Our approach consistently outperforms strong baselines across these datasets. Additionally, it reduces storage and runtime requirements by over an order of magnitude. These results show that high-quality retrieval can be achieved through LLM-driven text exploration without relying on static preprocessing or opaque vector representations.

## 1 Introduction

The advancement of Large Language Models (LLMs) has demonstrated remarkable capabilities in understanding and generating human-like text. However, their effectiveness is fundamentally constrained by their limited context window and memory capacity. Long context memory ability is crucial for LLMs' performance on long document Question Answering and multi-turn conversation tasks Lewis et al. (2021).

To address the memory and context limitations, *Retrieval-Augmented Generation* (RAG) has emerged as a prominent solution, with graph-based approaches showing particularly promising results. Graph-enhanced RAG systems (GraphRAG) leverage structural relationships between entities to enable more precise and comprehensive retrieval, capturing relational knowledge that traditional vector-based methods often miss. Recent empirical studies Edge et al. (2025); Fan et al. (2025) have demonstrated that integrating graph-based structures into RAG workflows could improve answer precision compared to vector-only retrieval methods, primarily due to the graph's ability to model complex relationships and dependencies between data points. Graph-based RAG systems excel at multi-hop reasoning tasks where answers require traversing multiple related concepts, making them particularly valuable for complex analytical queries that demand understanding of interconnected information.

Despite these advantages, graph-based RAG approaches face some challenges that limit their practical adoption. First, graph construction and maintenance usually require complex algorithms and specialized expertise, making their methods highly complicated and introduces new hyperparameters. These methods typically involve extensive hyperparameter tuning, including determining the optimal number of neighbors to retrieve, the depth of graph traversal (number of hops), and edge weighting schemes. Critically, these hyperparameters often need to vary dynamically based on the specific query characteristicssimple factual questions may require shallow exploration while complex analytical queries demand deeper graph traversal. This variability makes it challenging to identify universal parameter settings that work effectively across diverse query types.

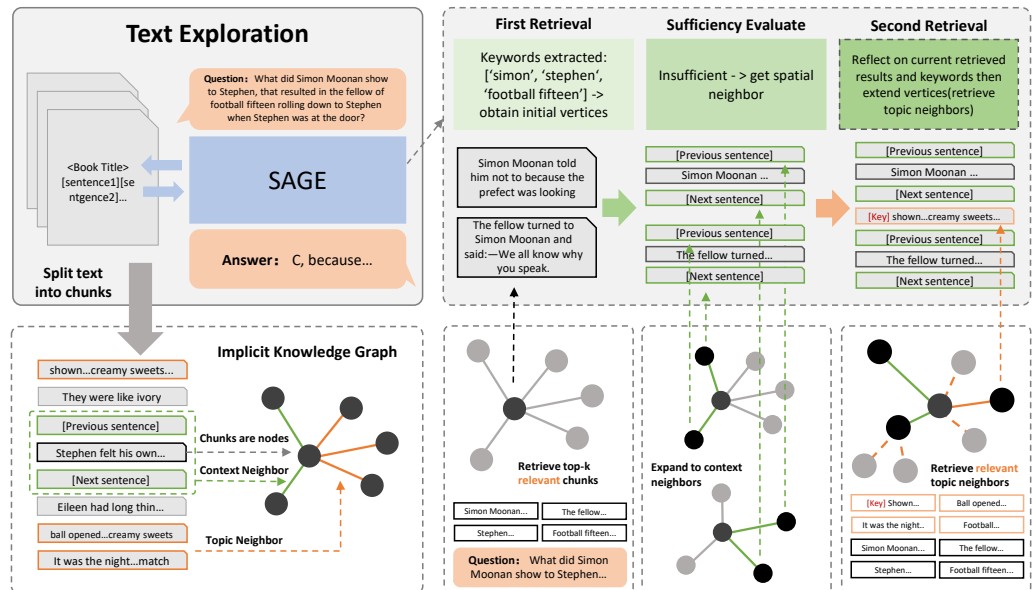

Figure 1: A demonstration of our method pipeline. Where "[key]" refers to the key evidence to answer the question.

Moreover, most existing Graph-based RAG methods rely on fixed edge types (such as character relationships, temporal connections, or predefined semantic relations) that are determined during the preprocessing phase. This static approach could limit the system's ability to adapt to novel query patterns or discover contextually relevant relationships that weren't anticipated during graph construction. Also, when answering domain-specific questions, the system cannot flexibly adjust its focus or exploration strategy based on the query context, potentially missing crucial connections that emerge only in specific question scenarios. Perhaps most critically, graph construction requires extensive preprocessing time that scales poorly with document length. Converting large documents into comprehensive knowledge graphs involves multiple passes of LLM-based entity extraction, relationship identification, and graph optimization, often requiring hours or even days for substantial corpora. This preprocessing bottleneck makes Graph-based RAG approaches impractical for dynamic environments where new documents are frequently added or updated, and creates a significant barrier to real-time applications.

Before the recent advances in LLM reasoning capabilities, these graph-based approaches represented the state-of-the-art for handling complex relational reasoning tasks. However, as LLMs have demonstrated substantially improved reasoning abilities, we can now design systems that leverage these enhanced cognitive capabilities to perform dynamic relationship discovery without explicit graph construction. This paradigm shift enables more flexible and adaptive retrieval strategies that can adjust their exploration patterns based on query context and iterative feedback.

We propose a dynamic implicit knowledge graph exploration method that conceptualizes documents as implicit graphs while avoiding explicit construction overhead. Our approach treats individual sentences as vertices and leverages LLM reasoning to discover contextually relevant edges on-demand. Through an iterative process of vertex discovery, sufficiency evaluation, and dynamic exploration, our method can adaptively traverse document relationships without predetermined graph structures. The system begins with keyword-based initial vertex selection, evaluates information sufficiency through objective vertex importance scoring, and dynamically expands the search scope through both spatial (contextual neighbor) and semantic relationship discovery.

Furthermore, recognizing the need to leverage LLM reasoning ability for reflective retrieval refinement, we adopt a text-level retrieval paradigm rather than embedding-based approaches. Unlike vector embeddings that compress semantic content into dense representations, text-level retrieval preserves complete information integrity and maintains full interpretability for LLMs. This transparency is crucial because embedding-based retrieval operates as a black boxLLMs cannot under-

stand how retrieval results were generated from their search queries, preventing them from effectively guiding subsequent exploration steps or adapting their search strategies based on retrieved content quality.

Our extensive experiments on challenging long-context QA benchmarks, including *NovelQA*(Wang et al., 2024a) and *Marathon*(Zhang et al., 2024), demonstrate that this embedding-free iterative exploration framework consistently outperforms strong embedding-based baselines while requiring significantly reduced computational and storage overhead. Our key contributions include:

1. An embedding-free retrieval framework that leverages LLM reasoning directly on text without requiring vector stores or dense indexing

2. A multi-round, LLM-driven retrieval process enabling effective iterative and adaptive text-level exploration with implicit graph traversal

3. Achieving state-of-the-art efficiency and accuracy on long-context QA benchmarks like NovelQA and Marathon while reducing runtime and storage requirements by orders of magnitude compared to other graph-based approaches

## 2 RELATED WORK

Existing approaches for retrieval-augmented long-context question answering can be categorized into four groups:

**Traditional Text Exploration.**    Early information retrieval systems, including BM25 (Robertson & Zaragoza, 2009) and keyword matching methods (Manning et al., 2008), perform retrieval directly at the text level by leveraging lexical overlap and term frequency signals. These methods are efficient and interpretable but struggle to capture semantic similarity, leading to degraded performance on reasoning queries.

**Embedding-Based Semantic Retrieval.**    Embedding models encode documents and queries into embedding vectors for semantic matching. Dense Passage Retrieval (DPR) (Karpukhin et al., 2020), Retrieval-Augmented Generation (RAG) (Lewis et al., 2021), and Contriever (Izacard et al., 2022a) exemplify this category, demonstrating improved retrieval relevance over lexical baselines. Recent modular retrieval frameworks, such as Atlas (Izacard et al., 2022b) and RePlug (Shi et al., 2023), further decouple retriever training from generators to enhance flexibility. While effective, embedding-based methods often act as black boxes, require significant storage for high-dimensional embeddings, and lack dynamic adaptability during inference. Moreover, researchers from Google also pointed out that embedding based retrieval have theoretical limitations Weller et al. (2025).

**Graph-Based and Memory-Augmented Exploration.**    To improve reasoning over multi-hop and long-range dependencies, graph-based retrieval approaches such as Graph-RAG (Edge et al., 2025), those using knowledge graphs (Wang et al., 2024b), and RAPTOR (Sarthi et al., 2024) incorporate entity and relational graphs to guide retrieval. Other novel systems like GraphReader (Li et al., 2024) build a graph from the text for an agent to autonomously explore. Memory-augmented systems, including Memory RAG (Qian et al., 2025), integrate token-level caches or explicit memory structures for factual consistency and longer context tracking. These methods enhance structured retrieval and multi-hop reasoning but introduce substantial preprocessing and storage overhead, limiting responsiveness in real-time applications.

**LLM-Augmented Iterative Exploration.**    Recent advances explore leveraging LLMs as active agents for retrieval and reasoning in an iterative manner. SelfRAG(Asai et al., 2023), Tool-Former (Schick et al., 2023a), ReAct (Yao et al., 2023), and LLM-as-a-Retriever (Shen et al., 2023) demonstrate how LLMs can guide multi-step retrieval and action planning for downstream tasks. These methods typically rely on embeddings or external tools for retrieval, maintaining partial dependence on embeddings or tool APIs. Unlike prior methods, our work introduces an *embedding-free, LLM-powered iterative text exploration* framework that directly retrieves and reasons over text without relying on vector embeddings or explicit graphs. This design achieves flexible, interpretable, and efficient retrieval while maintaining high accuracy on long-context QA benchmarks.

## 3 PRELIMINARIES

In this work, we consider the *Iterative Text Exploration (ITE)* problem that reframes Question-Answer matching as an *incremental decision process*: a LLM alternates between (i) proposing the most informative next chunk to inspect and (ii) evaluating whether the accumulated evidence is sufficient to answer the query. By leveraging the LLM's reasoning ability rather than static embeddings alone, ITE can dynamically surface query specific relations (e.g., selecting only business relevant friends) without prohibitive preprocessing.

Formally, we define the following notations:

- $\mathcal{Q}$: set of user queries (questions);

- $q \in \mathcal{Q}$: a single query;

- $\mathcal{C} = \{c_1, \ldots, c_N\}$: corpus of textual chunks (documents, paragraphs, or sentences) indexed by an external store;

- $Infer(q, x, \mathcal{L})$: a function that generates a natural-language answer $\hat{a}$ given $q$, context $x$ and a LLM $\mathcal{L}$;

- $T$: the maximum number of exploration iterations;

- $Score(q, \mathcal{C}, \hat{a})$: an evaluation function that returns a binary score from $\{0, 1\}$ to indicate if $\hat{a}$ correctly answers query $q$.

Then the *ITE* problem can be presented as: given a query $q$, corpus $\mathcal{C}$ and a LLM $\mathcal{L}$, the aim is to find an evidence $\mathcal{E} = \{e_1, \ldots, e_m\}$ satisfying

- $e_i \in \mathcal{C}$ for all $i$;

- $\hat{a} = Infer(q, \mathcal{E}, \mathcal{L})$ maximizes expected relevance to $q$ under *Score* while satisfying budget $T$.

## 4 METHODOLOGY

Recent advances in graph-based retrieval systems have demonstrated significant improvements in RAG performance through methods like GraphRAG Edge et al. (2025) and MiniRAG Fan et al. (2025). However, these approaches typically operate on statically constructed graphs, limiting their ability to leverage dynamic relations that emerge based on specific query contexts. For instance, when answering "introduce me one of your friends to do this business", an ideal system should identify friends related to the specific business domain rather than retrieving all friendship relations indiscriminately. Static graph construction would require extensive pre-classification of vertex properties, which becomes computationally prohibitive and may lack coverage for novel query patterns.

To address these limitations, we propose a dynamic implicit knowledge graph exploration method that leverages the reasoning capabilities of LLMs to discover contextually relevant relations on demand. We conceptualize the source document as an implicit knowledge graph $G = (V, E)$, where the vertices $V$ represent individual chunks, and the edges $E$ encode various semantic and contextual relationships. The proposed graph exploration process can be formalized as traversing an implicit knowledge graph where vertices represent concepts and edges encode semantic or logical relationships. Unlike static graph construction that requires $O(|V| + |E|)$ preprocessing complexity, our dynamic approach leverages LLM generative capabilities to identify relevant edges on-demand.

**Contextual Neighbor Edges**: The process exploit spatial proximity relationships by expanding around currently retrieved vertices. This breadth-first exploration captures semantically adjacent concepts through document structure, corresponding to traversing edges that connect spatially proximate sentences.

**Semantic Relation Edges**: When the currently retrieved vertices are not sufficient for answering the query, additional search $keys$ are generated based on retrieved vertex content, enabling depth-first exploration of semantic relationships. The LLM analyzes current vertices, the query, and previous search keys to identify complementary information needs, effectively discovering edges that represent logical or semantic relationships.

This iterative exploration strategy can be formalized as expanding the vertex set through two operations: a breadth-wise expansion to the spatial neighbors; and a depth-wise expansion to the dynamically discovered, semantic-relevant neighbors.

By avoiding explicit graph construction while achieving comparable information propagation benefits, our approach (Algorithm *ITE-QA*) maintains efficiency comparable to traditional retrieval methods while capturing complex multi-hop relationships that static approaches might miss.

This approach draws inspiration from human cognitive processes in handling long-context retrieval tasks. Humans typically cannot process entire documents at once; instead, they iteratively search for relevant keywords, assess the sufficiency of the retrieved information, and expand their search scope when necessary. By leveraging LLMs' ability to evaluate retrieval sufficiency, we simulate the fuzzy decision-making process that traditionally required human researchers.

To further enable LLMs to understand how their sufficiency decisions influence subsequent exploration steps, we adopt a keyword-based retrieval process rather than an embedding-based one. This choice helps preserve the interpretability of "relevance", allowing models to better reflect and control their retrieval behaviors.

As LLMs continue to improve, this text-level retrieval paradigm, which maintains information integrity, may prove superior to embedding-based methods that compress semantic content.

## 4.1 HANDLE COUNTING REQUESTS

**Algorithm 1:** *ITE-QA*

**Input:** $q, \mathcal{C}, T$
**Output:** $\hat{a}$
1 **if** *isCountingType(q)* **then**
2     $\hat{a} =$ Counting$(q, \mathcal{C}, \mathcal{L})$;
3 **else**
4     $\mathcal{E} := \emptyset, i := 0$;
5     **while** $i < T$ **do**
6       $keys :=$ GetKeys$(q, \mathcal{E})$;
7       $\mathcal{E} = \mathcal{E} \cup$ Retrieve$(keys, \mathcal{C})$;
8       **if** *isSufficient(q, $\mathcal{E}$)* **then**
9         break;
10       $i = i + 1$;
11     $\hat{a} :=$ Infer$(q, \mathcal{E}, \mathcal{L})$;
12 **return** $\hat{a}$;

**Algorithm 2:** *Importance$(q, \mathcal{E}, \mathcal{L}, n)$*

**Input:** $q, \mathcal{E}, \mathcal{L}$, n
**Output:** $score$
1 $sum := 0, i := 0$;
2 $a := Infer(q, \mathcal{E}, \mathcal{L})$;
3 **while** $i < n$ **do**
4     $\lambda := Uniform(0.3, 1]$;
5     $\mathcal{E}' := Disturb(\mathcal{E}, \lambda)$;
6     $a' := Infer(q, \mathcal{E}', \mathcal{L})$;
7     $sum = sum + CosSim(a, a')$;
8     $i = i + 1$;
9 $score := 1 - sum/n$;
10 **return** $score$;

Due to the limited performance of large language models (LLMs) in counting tasks, we detect and handle such queries via direct enumeration instead of iterative graph exploration as described by Algorithm 3. For all queries, we first prompt the LLM to determine whether or not it is counting type. Then for a counting query, we prompt the LLM to extract the target entity and its morphological variants (e.g., tenses, plurals) and then perform an exhaustive count of all occurrences. This approach circumvents the LLM's inherent weakness in precise numerical reasoning.

The decision to treat counting separately stems from well-documented challenges in the LLM community. For example, Kim & Schuster (2023) showed that even for those models pretrained on code, they struggle with entity tracking and counting in long contexts, often producing inconsistent results when maintaining state over multiple mentions. Similarly, Schick & Schütze (2021) highlighted that even smaller models can learn few-shot tasks with prompting, but tend to fail numerical reasoning without explicit supervision or fine-tuning.

Several methods have been proposed to improve LLMs' numerical and counting abilities. Chain-of-Thought prompting Wei et al. (2022) encourages step-by-step reasoning, but its effectiveness diminishes in long-document contexts where entity mentions are sparsely distributed. Toolformer Schick et al. (2023b) introduce mechanisms for integrating external tools into the generation process, but

also introduces systemic complexity and could not fully resolve the counting problem, particularly in unstructured text where counting is entangled with context understanding.

Simple statistical counting may also be insufficient to deal with counting questions, as it requires a certain amount of keyword extension ability based on the context. For example, to know how many times a character appears, we may want to search for both their full name and nickname.

Therefore, we categorize counting as an independent query type. This distinction is both methodologically necessary and practically vital.

## 4.2 ITERATIVE RETRIEVAL

For non-counting questions, we implement a relevance-based lexical vertex selection approach. The LLM generates contextually pertinent $keys$ derived from the query or concepts essential to its resolution (Prompt 1), and this keys serve as vertex selection criteria for identifying initial graph nodes.

---

**Algorithm 3:** $Counting(q, \mathcal{C}, \mathcal{L})$

**Input:** $q, \mathcal{C}, \mathcal{L}$
**Output:** $\hat{a}$
1   $keys := \text{GetKeysForCounting}(q)$;
2   $variant\_keys := \text{GetVariants}(keys)$;
3   $\hat{a} := \text{CountOccurrences}(keys, \mathcal{C})$;
4   **return** $\hat{a}$;

---

```
QUERY: {query}

RETRIEVED INFORMATION: {retrieved}

PROMPT: Extract proper keywords from
    both the query and its options.
```

Prompt 1: Prompt to generate keys based on the query and the retrieved information.

```
QUERY: {query}

RETRIEVED INFORMATION: {retrieved}

IMPORTANCE SCORE: {importance}

PROMPT: Given the retrieved
    information and the question
    with a calculated instructional
    importance of the information to
     answering the question from 0
    to 1, decide whether the
    information is enough to answer
    the question. If it is, say YES,
     if it is not, say NO.
```

Prompt 2: Prompt example to assess the sufficiency of retrieved information.

```
QUERY_WITH_OPTIONS: {
    query_with_options}

RETRIEVED INFORMATION: {retrieved}

PROMPT: Based on the retrieved data,
    make your own choose from the
    given options and explain why.
    If you are unsure of the correct
     answer, please make your best
    guess and still choose one of
    the options. YOU MUST OUTPUT the
     option you choose at the HEAD
    of your response.
```

Prompt 3: Prompt example to answer the query with the retrieved information.

Given the extracted $keys$, the retrieval process *Retrieve*$(keys, \mathcal{C})$ is called to return the chunks with top-k *relevance* scores to the given $keys$, where the *relevance* score between a chunk $c \in \mathcal{C}$ and a set of $keys$ is defined by

$$relevance(c, keys) = |\{w | w \in WordsOf(c) \cap keys\}|,$$

and *WordsOf* is the set of non-common words appearing in $c$.

At the end of each iteration, we assess whether the retrieved information suffices to answer the query, i.e. calling process *isSufficient*. Before diving into the implementations, notice that this kind of assessment usually faces two critical challenges:

1. LLMs lack intrinsic awareness of how much they actually utilize retrieved information during reasoning, potentially leading to overconfident sufficiency assessments based on hallucinated knowledge rather than retrieved content, and

2. the inherently subjective nature of sufficiency evaluation without objective guidance.

To address these limitations, we introduce an objective vertex importance metric that quantifies each retrieved chunk's contribution to the model's reasoning process. We formally define the importance of a text chunk $c \in \mathcal{C}$ to query $q$ as:

$$Importance(v, q; \theta) = -\mathbb{E}_{\lambda \sim U(0.3, 1]}[DisSim(c, q; \theta, \lambda)],$$

where $\theta$ denotes the language model parameters; $\lambda$ is a noise drawn from a uniform distribution from 0.3 to 1, and *DisSim* calculates a Monte Carlo estimated KL divergence between the posterior probability distributions of the LLM's output, comparing responses generated with the retrieved chunks versus those generated when the retrieved chunks are perturbed with varying levels of noise.

The perturbation noise is applied directly to the texts in various forms combined, including homoglyphs, common misspellings, OCR-related noise, and keystroke mistakes, with an overall probability of 0.7 for each word to be swapped with a noisy word. Detailed implementation of *DisSim* function is described in the B.1.

In actual implementation, we used Monte Carlo sampling across multiple noise levels to estimate this relationship, using cosine similarity between embedded outputs to capture semantic consistence, which results in Algorithm *Importance*.

After computing these objective "importance" scores, we incorporate them as supplementary information alongside the task prompt and retrieved chunks to guide the models assessment. These computed importance scores provide objective evidence to guide LLM sufficiency assessment (Prompt 2), addressing the hallucination pitfall where models may incorrectly assess sufficiency based on prior knowledge or hallucination rather than retrieved vertex information.

Finally, when the retrieved information is sufficient according to the assessment or the maximum number of iterations is achieved, *Infer* is performed by calling LLM to response to the given query (Prompt 3).

## 5 EXPERIMENTS

| Method | LLaMA 3.2 3b | LLaMA 3.1 8b | 70b | LLaMA 3.2 3b | LLaMA 3.1 8b | 70b |
|---|---|---|---|---|---|---|
| internal | 30.56 | 26.37 | 38.04 | 33.61 | 36.21 | 43.74 |
| vanilla | 36.57 | 56.02 | 44.55 | 54.84 | 61.34 | 66.29 |
| MiniRAG | 41.79 | 45.34 | 50.34 | 37.30 | 43.56 | 46.88 |
| Raptor | 50.37 | 52.56 | 58.96 | 49.50 | 56.46 | 65.17 |
| Ours | **55.54** | **63.29** | **71.27** | **55.67** | **61.51** | **68.46** |

(a) LLaMA on NovelQA (Left) and Marathon (Right)

| Model | ACC |
|---|---|
| selfRAG+gemma2:9b | 27.90 |
| GPT-3.5+RAG | 56.94 |
| Claude-v2:1 vanilla | 66.84 |
| GPT-4+RAG | 67.89 |
| Claude-3-Sonnet | 71.11 |
| GPT-4-0125-preview | 71.80 |
| Human performance | 97.00 |

(b) SOTA on NovelQA

| Model | ACC |
|---|---|
| Mistral-7B+RAG | 50.18 |
| Qwen-14B vanilla | 39.27 |
| Qwen-14B+RAG | 58.12 |
| Yi-chat-34B vanilla | 55.91 |
| Yi-chat-34B+RAG | 63.81 |
| GPT-4 vanilla | 78.59 |

(c) SOTA on Marathon

Table 1: Accuracy Performances on NovelQA and Marathon

In the experiments, we evaluate the performance of our proposed method and several State-of-the-Art(SoTA) RAG baselines, using LLaMA-3.1 (8B, 70B) and LLaMA-3.2 (1B, 3B) backbones. All models except 70b were tested on a single RTX 4070Ti SUPER GPU with 16GB VRAM. The 70b model was evaluated on a Mac Studio with 192 GB unified memory. We evaluated our method on the full datasets of two long-context multiple-choice QA benchmarks:

**NovelQA** Wang et al. (2024a) is a long-context QA benchmark with 2,305 multiple-choice questions from 88 novels, spanning aspects such as plot, character, and setting. The contexts could exceed 200,000 tokens, demanding long range multi-hop reasoning and memory ability. We evaluate models by accuracy on these multiple-choice tasks. This challenging dataset claims to have the longest average token length among all long-context datasets at the time of its publication.

**Marathon** Zhang et al. (2024) is a long-context QA benchmark with 1,530 multiple-choice questions covering six task types, including reasoning and retrieval. Each question has one correct answer and three verified distractors. Contexts average 100K characters, with some exceeding 260K, requiring deep reasoning under extreme length constraints.

We focused on **multiple-choice** benchmarks as they objectively evaluate retrieval quality without interference of generation quality and language style, as mentioned by Wang et al. (2024a).

| Interval | Avg Context Length | MiniRAG | | | | Raptor | | | | Our Method | | | |
|---|---|---|---|---|---|---|---|---|---|---|---|---|---|
| | | Prep. | Retr. | Gen. | Total | Prep. | Retr. | Gen. | Total | Prep. | Retr. | Gen. | Total |
| $<10^5$ | 79,028.343 | 348.630 | 6.564 | 1.511 | 356.705 | 108.741 | 0.031 | 4.600 | 113.372 | 0.004 | 0.774 | 2.725 | **3.503** |
| $(10^5, 5*10^5)$ | 170,635.733 | 648.515 | 7.757 | 1.721 | 657.993 | 208.254 | 0.042 | 4.906 | 213.203 | 0.007 | 0.975 | 2.984 | **3.966** |
| $(5*10^5, 10^6)$ | 752,523.312 | 2,390.854 | 6.193 | 4.202 | 2,401.249 | 759.523 | 0.116 | 6.475 | 766.114 | 0.017 | 1.684 | 6.485 | **8.186** |
| $(10^6, 2*10^6)$ | 1,346,647.368 | 4,564.582 | 6.520 | 4.189 | 4,575.291 | 1,268.484 | 0.123 | 6.321 | 1,274.929 | 0.030 | 2.451 | 6.523 | **9.004** |
| $>2*10^6$ | 4,103,032.667 | 19,337.624 | 6.849 | 4.518 | 19,348.990 | 6,423.175 | 0.181 | 6.344 | 6,429.700 | 0.085 | 6.047 | 12.546 | **18.678** |

Table 2: Average runtime of different methods across context-length intervals and processing phases (in seconds)

The evaluated RAG baselines and ablation studies include:

- **Internal**: Relies exclusively on the model's pre-trained parametric knowledge without any external document access. This baseline establishes the performance floor and quantifies the benefit of external knowledge retrieval, particularly relevant since NovelQA contains well-known literary works that may be present in training data.
- **Vanilla**: Directly concatenates the entire source document with the query as model input, leveraging the model's full context window without any retrieval mechanism. This approach tests the upper bound of long-context processing capabilities but is limited by context length constraints.
- **MiniRAG** Fan et al. (2025): Is a lightweight semantic-aware graph for retrieval, enabling small models to perform competitively with low memory and storage overhead.
- **RAPTOR** Sarthi et al. (2024): Applies recursive, tree-structured summarization for hierarchical retrieval, enhancing long-context reasoning with fewer tokens.

**Experiment Configurations.** The evaluation of our proposed method is performed with the following configurations:

- the number of vertices retrieved for the initial retrieval operation based on relevance (`recall_index`) is 25;
- the total number of spatial neighbors (preceding and succeding sentences) collected for the top 5 most relevant vertices (`neighbor_num`) is 20;
- the number of voters to answer the question given the same final retrieved results (`voter_num`) is 10; and
- is the maximum number of extension iteration in our iterative retrieval (`iter_max`) is 2.

For the baselines, we strictly follow the settings in the original works.

## 5.1 RESULTS AND ANALYSIS

| Dataset | Size (MB) | Method | Extra (MB) | Ratio |
|---|---|---|---|---|
| NovelQA | 72.4 | MiniRAG | 768.1 | 10.6x |
| | | Raptor | 1421.2 | 19.6x |
| | | Ours | 0.0 | 1.0x |
| Marathon | 146.1 | MiniRAG | 2437.4 | 16.7x |
| | | Raptor | 2862.6 | 19.6x |
| | | Ours | 0.0 | 1.0x |

Table 3: Storage comparison

| Param | Setting | Acc (%) |
|---|---|---|
| Voters(w/o importance) | 3 | 56.46 |
| | 5 | 57.25 |
| | *10* | 58.52 |
| Recall(w/o importance) | 20 | 58.56 |
| | *25* | 59.40 |
| | 30 | 58.69 |
| **Complete Method** | | **63.29** |

Table 4: Ablation study. Italic values match our full method.

Table 1 compares QA accuracy across retrieval strategies on NovelQA and Marathon.

In **NovelQA**, our method consistently outperforms baselines across all LLaMA scales. At the 70b scale, it achieves 71.27%, surpassing MiniRAG Raptor, Claude-3-Sonnet, and approaches GPT-4 (71.80%). Even with only 8b parameters, it exceeds GPT-3.5+RAG and approaches Claude-v2 (66.84%). These results demonstrate strong generalization under limited capacity and near parity

Table 5: Results of our method with various backbone on the novelQA

| Model | Overall Accuracy |
|---|---|
| gemma2:9b | 61.10% |
| deepseek-chat-v1 | 72.40% |
| qwen2.5 7b | 59.05% |
| qwen2.5 3b | 46.74% |

Table 6: Ablation study on `iter_max` hyperparameter (gemma2:9b)

| iter_max | Accuracy |
|---|---|
| 2 | 56.72% (1105/1948) |
| 3 | 56.65% (1061/1873) |
| 4 | 57.30% (1021/1782) |
| 5 | 58.10% (979/1685) |

with proprietary systems.

In **Marathon**, our method shows stable gains across scales, overtaking Raptor from 3b upward. At 70b, it reaches 68.46%, outperforming Raptor, MiniRAG, and all of the SOTA models except GPT-4. The trend indicates our robust scalability and long-context reasoning.

Table 4 presents an ablation study on NovelQA examining the impact of importance metrics alongside *voter_num* and *recall_index* parameters. Without importance guidance, all configurations achieve lower accuracy than our complete method. The results show that accuracy increases with more voters, while recall index exhibits an optimal value around 25, with both higher and lower values degrading performance.

Table 6 presents the ablation study on the effect of `iter_max` on the resulting accuracy on gemma2:9b model. With higher `iter_max` value, we obtain higher accuracy but increased time consumption for each retrieval.

As detailed in Table 5, our method maintains robust performance across diverse backbone LLMs, highlighting its strong generalizability.

### 5.1.1 RESPONSE TIME ANALYSIS

We evaluate response time using LLaMA-3.1:8b, dividing the QA process into **offline** and **online** phases. The **offline phase** includes *preparation time*, covering graph construction, chunking, and embedding-based indexing. The **online phase** comprises *retrieval time*, the latency of fetching relevant content, and *generation time*, which measures the duration from receiving the context to producing the final answer.

Table 2 presents a detailed breakdown of the average runtime for MiniRAG, Raptor, and our method across different context-length intervals and processing phases. Across all intervals, our method consistently demonstrates the lowest total latency, with total processing time ranging from only 3.5 to 18.7 seconds, while MiniRAG and Raptor exhibit significant increases up to 19,349.0 and 6,429.7 seconds respectively-primarily due to the overhead in the preparation phase.

Preparation time dominates the total cost for both MiniRAG and Raptor, particularly as the context length increases, reaching 19,337.6 seconds and 6,423.2 seconds in the >2e6 interval. In contrast, our method achieves near-zero preparation overhead (e.g., 0.085 seconds), as it does not require graph construction or embedding-based indexing. Retrieval time remains low and stable across all methods, while generation time increases moderately with context size but remains manageable.

To better capture per-query efficiency, we define the Average Query Time as:

$$\text{AvgQueryTime} = \frac{T_{\text{prepare}} + N \cdot (T_{\text{retrieve}} + T_{\text{gen}})}{N} \tag{1}$$

where $T_{\text{prepare}}$ is the one-time preprocessing cost, $T_{\text{retrieve}}$ and $T_{\text{gen}}$ are the per-query retrieval and generation times, and $N$ is the number of queries. This metric reflects the amortized latency per query.

### 5.1.2 EXTRA STORAGE ANALYSIS

Table 3 shows storage expansion ratios for MiniRAG, Raptor, and our method on NovelQA and Marathon datasets. MiniRAG increases data size by 10.6x and 16.7x respectively, while Raptor consistently expands storage by 19.6x across both datasets. Our method maintains a 1.0x ratio, storing

only original texts without additional data. The large overheads in MiniRAG and Raptor result from dense similarity matrices and graph-based indexing structures, which our method eliminates.

## 6 CONCLUSION

We presented SAGE, a dynamic implicit graph exploration framework that eliminates explicit graph construction in retrieval-augmented generation while preserving multi-hop reasoning. Our approach treats documents as implicit knowledge graphs with on-demand relationship discovery, addressing high preprocessing costs and static architectures of existing graph-based RAG systems.

We also designed a sufficiency-aware exploration mechanism with objective scoring, providing LLMs principled guidance for retrieval completeness to avoid hallucination-prone assessments.

This work demonstrates that high-quality retrieval can be achieved through LLM-driven text exploration without vector embeddings or explicit knowledge graphs, suggesting a paradigm shift toward more interpretable and adaptive retrieval architectures. The combination of dramatic efficiency improvements and competitive accuracy makes our approach particularly suitable for dynamic environments requiring minimal preprocessing overhead.

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

## A  DECLARATION OF LLMs USAGE

We used LLMs to polish our writing(grammar, correct usage of vocabulary, etc). But LLMs are not involved in more complicated research tasks, such as research ideation, retrieving related works, conducting experiments, etc.

## B  APPENDIX

This appendix provides comprehensive details on our importance score calculation methodology and presents empirical evidence of vector embedding limitations in retrieval tasks. We address two key aspects: (1) the mathematical foundation and approximation justification for our importance metric, and (2) systematic evaluation of embedding model performance across different query-answer relationship types.

### B.1  IMPORTANCE SCORE AS KL DIVERGENCE APPROXIMATION

#### B.1.1  THEORETICAL FOUNDATION

Our importance metric fundamentally measures how much a text chunk influences a language model's output distribution. We formally define the importance of a chunk $c$ to a query $q$ as the expected KL divergence between the model's output distributions when using the original chunk versus noise-perturbed versions:

$$\mathcal{I}mportance(c, q; \theta) = -\mathbb{E}_{\lambda \sim U(0.3,1]}[DisSim(c, q; \theta, \lambda)] \tag{2}$$
$$= -\mathbb{E}_{\lambda \sim U(0.3,1]}[D_{kl}(P(y \mid (c \oplus q))\|P(y \mid (\varepsilon_\lambda(c) \oplus q)))]$$

where:

- $c$ represents the text chunk being evaluated
- $q$ denotes the input query
- $\theta$ parameterizes the language model
- $\lambda$ controls the noise level (probability of character-level perturbation)
- $\oplus$ denotes text concatenation
- $\varepsilon_\lambda(c)$ applies noise to chunk $c$ with intensity $\lambda$
- $P(y \mid \cdot)$ represents the model's output probability distribution

The negative sign ensures that higher KL divergence (indicating greater distributional difference) corresponds to higher importance scores.

#### B.1.2  APPROXIMATION METHODOLOGY

Direct computation of Equation 2 is computationally intractable due to the need to evaluate probability distributions over the entire vocabulary space for each token position. We therefore employ the following approximation:

$$\mathcal{I}mp(c, q; \theta) \approx 1 - \text{AvgSim}(\theta(c \oplus q), \theta(\varepsilon_\lambda(c) \oplus q)) \tag{3}$$

#### B.1.3  MATHEMATICAL JUSTIFICATION

The validity of this approximation rests on several key assumptions about the language model's output distribution:

**Assumption 1: Distributional Form Consistency**  We assume that $P(y \mid \cdot)$ maintains a consistent functional form (e.g., multinomial with softmax normalization) across different inputs, with the primary variation occurring in the distribution's location parameters (means/modes) rather than scale parameters.

**Assumption 2: Temperature-Controlled Variance**  Under fixed temperature settings in the model's hyperparameter settings, the variance of output distributions remains relatively constant, making the mean the dominant factor in KL divergence calculations.

Under these assumptions, the KL divergence can be approximated as:

$$D_{kl}(P\|Q) \approx \frac{1}{2\sigma^2}\|\mu_P - \mu_Q\|^2 + O(\sigma^{-4})$$
$$\propto 1 - CosSim(\mu_P, \mu_Q) \qquad (4)$$

where $\mu_P$ and $\mu_Q$ represent the distribution means, $\sigma^2$ is the assumed constant variance, and the proportionality in Equation 4 holds when the vectors have similar magnitudes.

### B.1.4 IMPLEMENTATION DETAILS

Our Monte Carlo estimation procedure involves:

1. **Noise Generation**: We sample different noise levels following the distribution $\lambda_i \sim U(0.3, 1]$ for each chunk-query pair
2. **Text Perturbation**: Using the string-noise library[1], we apply character-level noise including substitutions, insertions, deletions, and transpositions
3. **Response Generation**: Generate responses from the language model for both original and perturbed inputs
4. **Similarity Computation**: Calculate cosine similarity between embedding representations of generated responses
5. **Aggregation**: Compute the average similarity across all noise levels: $\text{AvgSim} = \frac{1}{n}\sum_{i=1}^{n} CosSim(\mathbf{e}_{\text{orig}}, \mathbf{e}_{\text{noise}_i})$

The final importance score is then: $\mathcal{I}mp(c, q; \theta) = 1 - \text{AvgSim}$

## B.2 EMPIRICAL ANALYSIS OF EMBEDDING MODEL LIMITATIONS

### B.2.1 EXPERIMENTAL DESIGN

We conducted a systematic four-part evaluation to assess embedding model performance across different types of query-answer relationships:

**Dataset and Setup**  Our experiments utilized the NovelQA dataset, focusing on three question categories:

- **Character-sh**: Single-hop character identification questions
- **Relat-mh**: Multi-hop relationship reasoning questions
- **Span-mh**: Multi-hop span extraction questions

**Similarity Evaluation Framework**  For each query, we computed cosine similarity against four distinct text types:

1. **Q ANS**: Human-verified correct answers from NovelQA
2. **REL CLUE**: LLM-generated relevant contextual information that supports the correct answer without explicitly mentioning any of the answer choices and could be syntactically different from the query.
3. **IRR CLUE**: LLM-generated syntactically similar but irrelevant information that provides no discriminative value for answer selection
4. **RANDOM**: Randomly sampled passage sentences as baseline comparison

---

[1]https://github.com/dleemiller/string-noise

**Clue Generation Protocol** We employed Llama 3.1 8B for clue generation with carefully designed prompts and validated our clue generation by manually reviewing 30 randomly selected examples per category to ensure adherence to the generation criteria.

### B.2.2 RESULTS AND ANALYSIS

Table 7 presents comprehensive results across four state-of-the-art embedding models:

| Model | Category | Q ANS | REL CLUE | IRR CLUE | RANDOM |
|-------|----------|-------|----------|----------|--------|
| Mxbai-embed-large | Character-sh | 0.4326 | 0.5441 | **0.5978** | 0.3734 |
| | Relat-mh | **0.6890** | 0.5323 | 0.6010 | 0.3395 |
| | Span-mh | 0.4778 | 0.4242 | **0.5596** | 0.3184 |
| NV-Embed-v1 | Character-sh | 0.3333 | 0.2786 | **0.4397** | 0.2952 |
| | Relat-mh | **0.5487** | 0.3192 | 0.4842 | 0.3078 |
| | Span-mh | 0.4247 | 0.2196 | **0.4696** | 0.2436 |
| Jina-v3-text-matching | Character-sh | 0.4855 | 0.5947 | **0.6584** | 0.4492 |
| | Relat-mh | **0.7330** | 0.6043 | 0.6631 | 0.4185 |
| | Span-mh | 0.5488 | 0.4843 | **0.6332** | 0.3854 |
| Jina-v3-retrieval | Character-sh | 0.0002 | **0.3210** | 0.2205 | 0.0837 |
| | Relat-mh | **0.4050** | 0.3483 | 0.2417 | 0.0410 |
| | Span-mh | 0.1744 | 0.1828 | **0.2008** | 0.0042 |

Table 7: Cosine similarity scores between query embeddings and different content types across embedding models and question categories. Bold values indicate highest similarity within each row. Higher scores for IRR CLUE than REL CLUE or Q ANS indicate problematic retrieval behavior.

### B.2.3 KEY FINDINGS

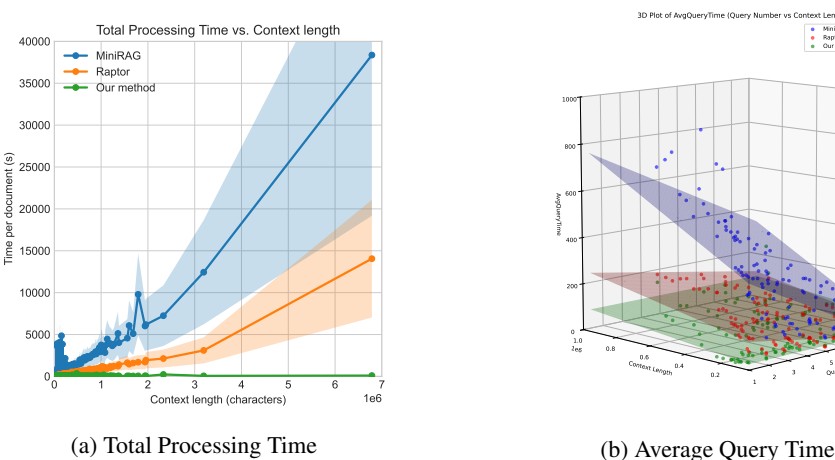

(a) Total Processing Time

(b) Average Query Time (3D)

Figure 2: Breakdown comparison of total time consumption of our method against Raptor and Mini-RAG across different stages.

**Syntactic Bias Over Semantic Relevance** Three of four models (Mxbai-embed-large, NV-Embed-v1, Jina-v3-text-matching) consistently assign higher similarity scores to irrelevant clues than to actual answers or relevant information. This indicates a concerning tendency to prioritize syntactic similarity over semantic relevance.

**Model-Specific Behaviors**

- **Jina-v3-retrieval**: Shows the most promising behavior by generally preferring relevant content, though still fails in span-based reasoning tasks

- **General-purpose embeddings**: Models not specifically fine-tuned for retrieval exhibit more severe syntactic bias

- **Question type sensitivity**: Multi-hop reasoning questions (Relat-mh) generally show better performance across models

**Proximity to Random Performance**   Several model-category combinations approach random baseline performance, indicating fundamental limitations in distinguishing meaningful content relationships.

### B.2.4   IMPLICATIONS FOR RETRIEVAL SYSTEMS

These findings highlight critical challenges for embedding-based retrieval:

1. **Test-time Uncertainty**: Without knowledge of question category or content type, it is impossible to predict when an embedding model will fail

2. **Query Expansion Risks**: LLM-based query expansion may introduce additional noise without guaranteeing improved retrieval performance, particularly when the expanding model lacks knowledge of the target embedding space characteristics

### B.3   DETAILED TIME CONSUMPTION COMPARISON AND ACCURACY ANALYSIS

Besides the time consumption analysis performed in our Experiments section, we also conducted some additional analysis on the time consumption by stages and the prepare time consumption when using different LM context lengths of our method versus other SOTA retrieval methods like miniRAG and RAPTOR, results are shown in Figure 2 and Figure 4. Some additional analysis of our accuracy on NovelQA using different context lengths and across different question types are also conducted, see Figure 3 and Figure 4.

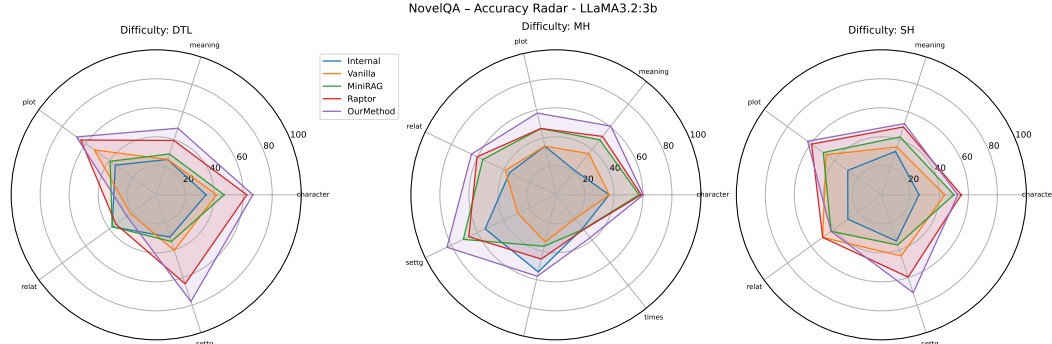

Figure 3: Accuracy comparison between our method and other methods across different types of questions. Where MH denotes multi-hop reasoning questions, SH denotes single-hop reasoning questions, and DTL denotes detail questions without the need to perform any reasoning. Our method shows superior performance in all threee types, especially in Multi-hop reasoning questions.

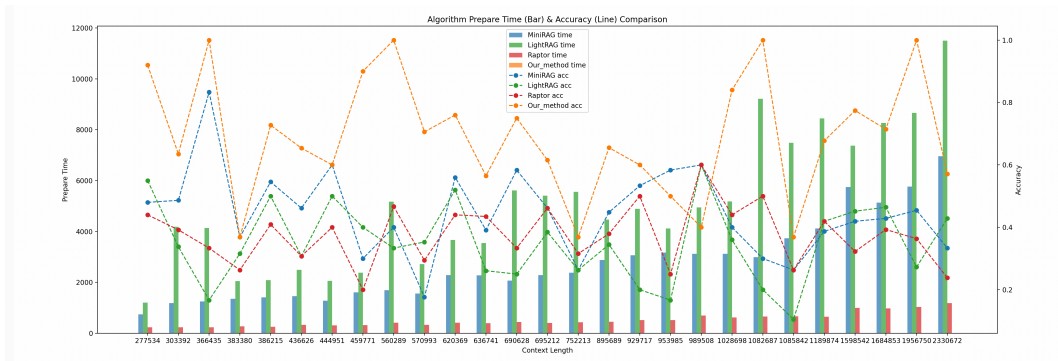

Figure 4: Time consumption and accuracy vs LM context length used results. Our method shows minimal preparation time(not even visible on the chart), while almost always out-performing other methods in accuracy across most context lengths tested.

