# OpenReview forum: "SAGE: Sufficiency-Aware Implicit Graph Exploration for Long Context Reasoning"
_ICLR.cc/2026/Conference — Submitted to ICLR 2026_

### Official Review · Reviewer_pfgd · 2025-10-15

**Soundness:** 2
**Presentation:** 2
**Contribution:** 2
**Rating:** 4
**Confidence:** 4

**Summary:**

The paper proposes SAGE (Sufficiency-Aware implicit Graph Exploration), a retrieval-augmented framework for long-context reasoning that avoids explicit graph construction and dense vector indexing. Instead, SAGE treats documents as an implicit graph of chunks and lets an LLM iteratively (i) generate query-conditioned keywords, (ii) retrieve candidate chunks lexically, (iii) assess sufficiency using an objective importance score that measures each chunk’s causal impact on the answer distribution via noise perturbations, and (iv) expand along spatial (context neighbors) and semantic (topic neighbors) edges only when evidence is insufficient. On NovelQA and Marathon, SAGE reports higher accuracy than MiniRAG/RAPTOR across multiple LLaMA scales while cutting storage and preparation time by orders of magnitude, since it stores only the original text and skips any pre-built graphs/embeddings.

**Strengths:**

1. Pre-built graphs are costly and rigid, and embedding-based methods add storage overhead and reduce interpretability in long-context settings.
2. The method’s combination of implicit graph exploration, keyword-based retrieval, and sufficiency-aware stopping forms a clear and interpretable pipeline.
3. The importance score, based on light perturbation and KL evaluation, helps prevent premature stopping and reduces hallucination.
4. It is highly efficient: almost zero preprocessing, only original text stored (≈1×), yet it matches or outperforms MiniRAG/RAPTOR, especially with 8B/70B models.
5. Ablation studies show the benefit of each component (voters, expansion, importance signal), and performance is robust across context lengths.

**Weaknesses:**

1. Lexical retrieval ceiling: While interpretability is a plus, relying on keyword overlap risks recall issues for paraphrased, non-lexically aligned evidence; a hybrid (lexical + light semantic) comparison would strengthen claims of “embedding-free superiority.”
2. Benchmark scope: Only multiple-choice QA (NovelQA/Marathon). It would be valuable to add free-form QA and multi-document synthesis (e.g., GovReports, Qasper, LongFact) to test robustness beyond MCQ cueing effects.
3. Importance metric assumptions: The KL→cosine approximation assumes near-constant variance/softmax temperature; an empirical sensitivity study (temperature, noise distributions, #samples) would bolster soundness.
4. Counting branch: Counting is handled via a special pipeline. This is practical, but it highlights that SAGE still needs task-specific forks; a unified controller that chooses among skills/tools could generalize better.
5. Comparisons vs. stronger agentic baselines: Given the “LLM-as-retriever/agent” lineage, adding ReAct- or Toolformer-style agentic retrieval baselines (without embeddings) would clarify where SAGE’s sufficiency gating provides the most benefit. The set of compared methods in the paper is currently too limited.

**Questions:**

See weaknesses.

---

> ### Author Response · Authors · 2025-11-21
> **Reply to Reviewer pfgd**
>
> *1. Regarding the lexical retrieval ceiling.*
>
> We agree that "embedding-free superiority" is a strong claim, but we provide new justifications.
> * Recent work from Google DeepMind ("On the theoretical limitations of embedding-based retrieval, 2025") reveals fundamental limitations in embedding-based retrieval.
> * Our analysis in Appendix B.2 provides some evidence toward using "embedding-free" over "embedding-based": many SOTA embedding models (e.g., Mxbai-embed-large) suffer from "syntactic bias," scoring irrelevant but lexically similar clues higher than the actual answer.
> * Our LLM-guided lexical exploration avoids this pitfall. Furthermore, our method can actually retrieve non-lexically aligned evidence by dynamically extending search words based on retrieval results.
>
> *2. Regarding the benchmark scope (MCQ only).*
>
> We chose MCQ benchmarks (NovelQA, Marathon) for their challenging long context (as far as the author's knowledge, NovelQA has the longest context length among all the QA benchmarks) and their ability to provide objective evaluation of retrieval quality without the confounding variable of generation syntactic style. We believe these benchmarks are a sufficient test for the paper's core claims about long-context retrieval.
>
> *3. Regarding the counting branch "fork".*
>
> We thank you for the advice. This is an explicit design choice where we view the system as using a form of "tool use". The system intelligently identifies a task-specific failure mode (LLM counting) and routes it to a specialized module, which is more robust than a single unified controller.
>
> *5. Regarding comparisons vs. stronger agentic baselines.*
>
> We have added results for SelfRAG, a prominent agentic RAG method, on the NovelQA dataset in Table 1(b). In terms of the graph-based RAG methods, RAPTOR and miniRAG are two of the prominent and powerful hierarchical and graph-based RAG methods we selected.

---

### Official Review · Reviewer_i7LV · 2025-10-27

**Soundness:** 2
**Presentation:** 1
**Contribution:** 2
**Rating:** 2
**Confidence:** 5

**Summary:**

This paper introduces SAGE, a framework for dynamic implicit graph exploration. SAGE aims to preserve multi-hop reasoning capabilities in RAG by treating documents as implicit knowledge graphs where relationships are discovered on-demand, thereby eliminating the need for explicit graph construction.

**Strengths:**

N/A

**Weaknesses:**

1. The ad-hoc handling of "counting" problems, as described in Section 4.1, seems superfluous and undermines the method's generality. The technique employed for this specific task is highly intuitive and appears to offer minimal technical contribution.

2. The paper's overall presentation is poor, suffering from unclear writing and numerous formatting errors, which makes the methodology difficult to follow. For instance:

    a. A "DisSim" function is mentioned on line 329, with a reference to Appendix B.1 for implementation details. However, Appendix B.1 describes an "Imp()" function, and "DisSim" is never mentioned again.

    b. Algorithm 3 is included in the paper but is never referenced or explained in the main text.

    c. There are multiple distracting formatting errors, such as missing parentheses for citations (e.g., lines 112, 194) and improper spacing around parentheses (e.g., line 412).

4. The related work section is insufficient and misses several critical citations. The paper fails to discuss or position itself against highly relevant recent work, such as:
```
@inproceedings{wang2024knowledge,
title={Knowledge graph prompting for multi-document question answering},
author={Wang, Yu and Lipka, Nedim and Rossi, Ryan A and Siu, Alexa and Zhang, Ruiyi and Derr, Tyler},
booktitle={Proceedings of the AAAI conference on artificial intelligence},
volume={38},
number={17},
pages={19206--19214},
year={2024}
}
@article{li2024graphreader,
title={Graphreader: Building graph-based agent to enhance long-context abilities of large language models},
author={Li, Shilong and He, Yancheng and Guo, Hangyu and Bu, Xingyuan and Bai, Ge and Liu, Jie and Liu, Jiaheng and Qu, Xingwei and Li, Yangguang and Ouyang, Wanli and others},
journal={EMNLP},
year={2024}
}
```

**Questions:**

N/A

---

> ### Author Response · Authors · 2025-11-21
> **Reply to Reviewer i7LV**
>
> *1. Regarding the "ad-hoc" counting method.*
>
> We agree that the counting module (Section 4.1) is a practical component. It is a deliberate design choice to handle a well-known failure mode of smaller-scale LLMs.
> * We did not handpick counting problems; rather, they are identified by LLMs, and the counting pipeline is part of the entire method.
> * We believe this is a feature that makes SAGE robust, while our core technical contribution remains the novel implicit graph exploration (Section 4.2).
>
> *2. Regarding presentation and errors.*
>
> We sincerely apologize for the errors and have thoroughly revised the manuscript.
> * **'DisSim' vs 'Imp':** We have rewritten Appendix B.1 to provide a clear theoretical justification for the Imp() function as a cosine-based approximation of KL divergence. The connection is now explicit.
> * **Algorithm 3:** We now explicitly reference Algorithm 3 in Section 4.1.
> * **Formatting:** We have proofread the paper and fixed the cited formatting errors (e.g., lines 112, 194, 412).
>
> *3. Regarding insufficient related work.*
>
> Thank you for these highly relevant citations. We have added both *wang2024knowledge* and *li2024graphreader* to our Related Work (Section 2) and discussed their connection to our work.

---

### Official Review · Reviewer_sFCj · 2025-10-30

**Soundness:** 3
**Presentation:** 2
**Contribution:** 3
**Rating:** 4
**Confidence:** 4

**Summary:**

The paper proposes SAGE, a sufficiency-aware implicit graph exploration framework designed to enhance long-context reasoning for Large Language Models (LLMs).

Instead of relying on explicit graph construction or vector embeddings, SAGE enables dynamic implicit graph traversal over text chunks using LLM reasoning capabilities. The framework introduces a sufficiency evaluation mechanism to decide when enough information has been retrieved, guided by an objective importance score that measures how much a text chunk influences the model’s reasoning.

Experiments on NovelQA and Marathon benchmarks show that SAGE achieves SOTA accuracy and efficiency, outperforming graph-based methods like MiniRAG and Raptor, while reducing preprocessing time and storage by orders of magnitude. The authors argue that this embedding-free, reflective retrieval paradigm marks a paradigm shift toward more interpretable and adaptive long-context retrieval systems.

**Strengths:**

- SAGE eliminates the need for embedding and graph construction, significantly enhancing retrieval efficiency for long-context QA.
- SAGE introduces an objective, effective, and novel method for evaluating the importance of evidence in RAG.

**Weaknesses:**

- The paper's description of the methodology lacks clarity, making it difficult to correlate the main text, the algorithm, and the pipeline in Figure 1.
- The experiments are insufficient. The paper compares too few baselines and does not include a comparison with the GraphRAG paradigm mentioned at the beginning.
- The ablation study design has problems: firstly, it lacks experiments on the impact of iteration rounds $T$ on results; secondly, it fails to provide comparisons between with and without importance scores under non-default values of voter_num and recall_index.
- Experiments are limited to LLaMA family models. The generalization to other LLMs is not verified.

**Questions:**

Refer to Weaknesses.

---

> ### Author Response · Authors · 2025-11-21
> **Reply to Reviewer sFCj**
>
> *1. Regarding clarity of the methodology.*
>
> We apologize for the lack of clarity. We have revised parts of Section 4 to better connect the text, Figure 1, and the algorithms.
> * We now explicitly reference Algorithm 3 in Section 4.1 when describing the counting pipeline.
> * We have improved the language in Section 4.2 to clearer describe the iterative retrieval and sufficiency evaluation.
>
> *2. Regarding insufficient baselines (GraphRAG).*
>
> We have updated our Related Work (Section 2) to position our work against recent iterative agent-based RAG paradigms like SelfRAG. We respectfully argue that our baseline comparison is appropriate because we compared against Raptor and MiniRAG, which are recent SOTA hierarchical and graph-based RAG methods.
>
> *3. Regarding problems in the ablation study.*
>
> We thank the reviewer for pointing this out and have clarified Table 4. The results with Voters=10 and Recall=25 utilize the default values (the exact same settings as our method), except they lack importance guidance. Additionally, we added an ablation study on the effect of increasing the `iter_max` parameter (T) in Table 6.
>
> *4. Regarding experiments limited to LLaMA.*
>
> We have addressed this by adding experiments on popular open-source models. As noted in the General Response and Table 5, our method demonstrates strong generalizability across Gemma, DeepSeek, and Qwen models.

---

### Official Review · Reviewer_CQkY · 2025-11-01

**Soundness:** 2
**Presentation:** 3
**Contribution:** 2
**Rating:** 2
**Confidence:** 4

**Summary:**

This paper proposes an LLM-based RAG approach that iteratively retrieve sentences. The major motivation is to treat sentence as retrieval units and perform graph-like search trajectory without the need of pre-building a graph. Specifically, it identifies keywords in each turn, retrieve sentences by keyword-based matching, and evaluate evidence importance by measuring the effects of it on the LLM's output. Experiments are mainly comparing with some graph-based baselines that show the reduced offline cost and higher performance.

**Strengths:**

- This paper is inspired by graph-based approaches while proposes a method that mimic a similar behavior without graph construction. This work is thus well motivated
- The proposed objective importance metric is interesting and seems to be effective. It worth further studying and could be applied in different cases as well

**Weaknesses:**

- The methodology is essentially two unrelated parts, one on counting queries and another one on the retrieval algorithm. This makes the paper lose a focused contribution, and more essentially, the experiments also do not show how the two parts contribute to the performance
- The selection of baselines is insufficient. For example, there are quite a number of graph RAG approaches and also some prompt-based iterative RAG methods without graph.
- Because the method does not involve training, it would be expected to evaluate on more than just open-source models like Llama.

**Questions:**

Please see weaknesses above

---

> ### Author Response · Authors · 2025-11-21
> **Reply to reviewer CQkY**
>
> *1. Regarding the methodology (two parts) and their contribution.*
>
> We thank the reviewer for this observation. Our methodology is indeed composed of two main branches: the "counting" module (Section 4.1) and the core SAGE retrieval algorithm.
>
> * **Justification for the Counting Branch:** This is a deliberate and practical design choice to handle a well-documented failure mode of LLMs (long-context counting), especially for the smaller-scale models (3b, 7b, 8b) we are using. This ensures system robustness but is not our major contribution.
> * **Contribution of SAGE:** Our core contribution is the SAGE retrieval paradigm, which focuses on more general types of questions. To demonstrate this, we added a per-type accuracy breakdown in Appendix B.3 (Figure 3). These radar charts show our method significantly outperforms baselines on Multi-Hop (MH) and Detail (DTL) questions, demonstrating the effectiveness of our implicit graph exploration.
>
> *2. Regarding the selection of baselines.*
>
> We appreciate this suggestion. We have expanded our Related Work (Section 2) to include several recent graph RAG and agent-based methods. While some methods lacked a ready-to-use code base, we successfully added a test on the SelfRAG method on the NovelQA dataset (Table 1(b)).
>
> *3. Regarding evaluation on non-Llama models.*
>
> Thank you for your suggestions. As detailed in the **General Response**, we have run new experiments on a diverse set of backbone models. The new results (Table 5) confirm that SAGE's benefits are not limited to the Llama family.

---

> > ### Comment · Reviewer_CQkY · 2025-11-26
> >
> > Thanks for the explanation. However, I still think the paper would benefit from a more comprehensive comparison on more baselines and across different models. The current added results (only SelfRAG) and Table 5 which only shows proposed method + different LLMs but no comparison with any baselines do not provide much valuable information to help understand the method.
> >
> > Also, for my W1, I totally understand the motivation of specific handling of counting problem, but that dilutes the focus of the paper and also lacks rigorous experiments. I checked the per-type results in the appendix, but the types are not aligned with whether or not it is a counting problem. I still think the experiments should explicitly rule out the benefits of the counting branch if the other part is the major claim of the paper.

---

### Author Response · Authors · 2025-11-21
**Reply to all reviewers**

We sincerely thank the reviewers for their constructive and insightful feedback. We have uploaded a revised manuscript addressing these points, with major changes highlighted in blue.

Based on the collective feedback, we have made the following major improvements to the manuscript:

* **Expanded Model Evaluation:** To address concerns regarding generalizability beyond the Llama family, we conducted new experiments on a diverse set of backbone models. As shown in the new Table 5, our method achieves high accuracy with gemma2:9b (61.10%), deepseek-chat-v1 (72.40%), and qwen2.5 7b (59.05%).

* **New Baselines and Related Work:** We have expanded our Related Work (Section 2) to include a recent agent-based method. We added a test on the SelfRAG method on the NovelQA dataset, with results reported in Table 1(b). We also integrated specific citations (e.g., *wang2024knowledge*, *li2024graphreader*) suggested by the reviewers.

* **Expanded Ablation Study:** We have added an ablation study on the effect of `iter_max` parameter to illustrate the effectiveness of our iterative retrieval strategy.

* **Clarification of Methodology:** We have revised the Methods section to better connect the text, Figure 1, and the algorithms. We have shown more explicit connections from the formulas in the Methods section to the ones described in the Appendix.

* **Minor Revisions on Grammar:** We have made some minor modifications to the syntax and grammar, and rephrased some sentences to make our paper easier to follow.

---

### Meta-Review · Area_Chair_XKeY · 2026-01-02

**Summary:**

This paper proposes an iterative RAG approach for LLMs, specifically for the long-context scenario. It treats sentences as retrieval units and perform iterative search of neighbor chunks until sufficient evidence has been gathered. The design avoid explicitly constructing the graph to achieve efficiency and allow for dynamicity.

The reviewers are concerned about the effectiveness of the proposed method (w.r.t. the proposed "counting module"), the lack of empirical evidence of performance gains w.r.t. different model families, and more comprehensive baselines, lack of ablation studies, and unclear technical descriptions.

The rebuttal partially addressed concerns, leaving some important points unresolved (e.g., the impact of the "counting module", the generalization across model families, and limited empirical evidence).

**Reviewer Concerns:**

# Reviewer CQkY

### Addressed:
* (Partially) additional baseline (SelfRAG) on NovelQA

### Unaddressed:
* Unclear contribution of the standalone counting module: this module is not related to the main contribution of SAGE and is a hand-tailored solution for a specific LLM failure mode. There lacks ablation to reveal the gains from the main SAGE module
* Results on more model families: the additional results on different model families do not carry much meaning due to lack of baseline performance

# Reviewer sFCj

### Addressed:
* (Partially) additional baseline. See above
* (Partially) ablation on T. There lacks results on other ablation parameters pointed out by the reviewer

### Unaddressed:
* Results beyond Llama. See above.


# Reviewer i7LV

### Addressed
* Presentation becomes better after incorporating reviewer's suggestions.
* More comprehensive related work

### Unaddressed
* Unclear contribution regarding counting module. See above.

# Reviewer pfgd

### Addressed
* (Partially) additional baseline. See above

### Unaddressed
* Importance metric assumptions: no response provided
* Lexical retrieval ceiling: while embedding-based retrieval has its limitation, it does not fully justify that a pure lexical retrieval method is the best. The reviewer's suggestion on a hybrid approach make conceptual sense, and it would be interesting to see a discussion on the limitation of the current pure lexical-based approach
* Limited benchmarks: I agree that the paper would be more convincing with a more diverse set of benchmarks
* Counting module. See above.

**Reviewer Scores:**

The revised version has improved presentation and included some additional experimental results. I would expect some reviewer(s) to increase their scores. However, since some critical concerns still remain unresolved, the final scores may not reach the acceptance bar.

---

### Decision · Program_Chairs · 2026-01-26

Reject